# Antioxidant Guided Fractionation of Blackberry Polyphenols Show Synergistic Role of Catechins and Ellagitannins

**DOI:** 10.3390/molecules28041933

**Published:** 2023-02-17

**Authors:** Katerina Tzima, Gontorn Putsakum, Dilip K. Rai

**Affiliations:** Teagasc Food Research Centre, Ashtown, D15KN3K Dublin, Ireland

**Keywords:** blackberries, flash chromatography fractionation, antioxidant indices, LC-MS/MS, phenolics, ellagitannins, anthocyanins, catechins

## Abstract

In the present study, blackberry extract was prepared using a previously optimized solid–liquid extraction method in 70% aqueous acetone aimed at the recovery of its principal phenolics. Subsequently, 0.5 g of freeze-dried extract was subjected to flash chromatography fractionation, which was conducted on a C18 column using a binary solvent system of water and methanol at 10 mL/min. The total phenolic content (TPC), 2,2-diphenyl-1-picrylhydrazyl (DPPH), and ferric reducing antioxidant power (FRAP) activities of the obtained 42 flash fractions were determined, and a strong positive correlation (*r* ≥ 0.986) was exhibited among them. Furthermore, the graph of the antioxidant indices of the flash fractions resembled the flash chromatogram, suggesting a good correlation among the compounds within the chromatographic peaks and the antioxidant indices. LC-MS/MS identified as many 28 phenolics, including cinnamtannin A2 reported for the first time in blackberries. This study further established the role of dominant anthocyanins (cyanidin-3-*O*-glucoside and cyanidin-3-*O*-rutinoside), but uniquely those of ellagitannins and catechins on the antioxidant capacity of blackberries.

## 1. Introduction

Berries of the *Rubus* genus, including blackberries, have attracted considerable scientific interest due to the presence of potent bioactive compounds, mainly anthocyanins and ellagitannins [1,2]. It is not only the high content of anthocyanins and ellagitannins in blackberries that is related to their high antioxidant capacity but also other phenolic compounds can be significant contributors to their antioxidant potential [3]. For instance, Gong et al. [4] had shown that blackberries possess fifteen potent antioxidants that included anthocyanins and other phenolic compounds, either in free or bound forms, after examining a number of different blackberry varieties [4]. On the other hand, Sariburun et al. [5] also found that the antioxidant activities of blackberries were highly correlated (0.93 ≤ r ≤ 0.99) with their anthocyanin contents, while the total flavonoid content was relatively less correlated with antioxidant activity (0.85 ≤ r ≤ 0.89). 

Various separation methods to enrich blackberry polyphenols and link to their antioxidant activity have been reported. Acosta et al. [6] separated blackberry polyphenols using membrane filtration of different molecular weight cut off (MWCO) sizes (1–150 kDa), where the authors found the membrane with MWCO of 2 kDa as the best for the fractionation of anthocyanins. Ghosh et al. [7] also applied membrane filtration sequentially (50 kDa followed by 300 Da nanofiltration) to successfully concentrate antioxidant anthocyanins from Indian blackberry (*Syzygium cumini*) juice. Elisia et al. [8], on the other hand, applied gel filtration to obtain an anthocyanin-enriched extract of blackberry (*Rubus fruticosus* sp). The total anthocyanin content following gel filtration was 20-fold higher than the original crude extract, while the antioxidant activity was enhanced by 7-fold for the anthocyanin-enriched fraction. Sánchez-Velázquez et al. [9] used column chromatography employing different solvents and adsorbent and ion-exchange resins (i.e., Amberlite XAD-7 resin followed by Sephadex LH-20 resin) to concentrate ellagitannins from Mexican wild blackberries. Cho et al. [10] applied semi-preparative HPLC to fractionate blackberry flavonols and interestingly showed that the flavonols had a minor (<5%) contribution toward the total antioxidant capacity in blackberries. In the present study, flash chromatography is employed for the first time to fractionate blackberry polyphenols and associate them to their antioxidant activity. For decades, flash chromatography (FLC) has been used for the separation of compounds of interest as it is a powerful, versatile, and scalable technique [11]. Antioxidant-guided identification constitutes a method that is based on the chromatographic fractionation of an extract, and the subsequent selection of the most antioxidant-active fractions owing to the presence of certain potent antioxidant compounds [12]. Upon selection of the most potent fractions, this purification method can facilitate their further isolation, even in the cases of extracts with low concentrations of potent antioxidants. The isolated fractions can then be used to confirm the identity and safety of antioxidant constituents for utilization as food additives [13]. Hence, phenolic-rich extracts could be generated inexpensively, and their utilization and downstream purification could increase the sustainability and cost effectiveness of the fruit industry, especially in the case of fruit waste products.

Therefore, the focus of this study was the assessment of a systematic fractionation approach employing flash chromatography to determine the most potent fractions of blackberry extract due to the presence of certain phenolic compounds. Subsequently, this study aimed to generate a method that could be potentially used after employing food-compatible solvents, such as ethanol or acetone, as a further purification step for isolating bioactives for food applications or fractionating compounds for the phyto-pharmaceutical industry. To achieve the objectives, three different chemical assays were employed to determine the most antioxidant-active fractions of a phenolic rich blackberry extract, including Folin-Ciocalteu (F-C), 2,2-diphenyl-1-picrylhydrazyl (DPPH), and ferric reducing antioxidant power (FRAP). The identification and quantification of the compounds responsible for the exerted antioxidant effect were performed using liquid chromatography tandem mass spectrometry techniques. To the best of our knowledge, this is the first study assessing the antioxidant-guided fractionation of blackberry polyphenols.

## 2. Results and Discussion

### 2.1. Antioxidant Indices of Phenolic-Rich Fractions of Blackberries

Automated FLC for the purification of phenolics has been successfully utilized for the identification of antioxidant activities attributable to either individual compounds or to synergistic effects among a group of compounds [14]. For example, the fractionation of sour orange using automated FLC and the evaluation of its phenolic rich fractions demonstrated that the presence of biological activity was potentially attained due to the synergism among nine phenolic constituents in the extracts [15]. 

In the present study, after FLC fractionation of a phenolic rich blackberry extract and through assessment of their TPC, DPPH, and FRAP indices, the fractions possessing the highest indices were subjected to LC-MS/MS analysis. The FLC separation of the extract produced three distinctive peak-regions as illustrated in its flash chromatogram (Figure 1). The graph of antioxidant indices (Figure 2) that was obtained after scrutinizing the 42 collected flash fractions followed the same trend with that of the flash chromatogram, exhibiting a satisfactory correlation with its chromatographic peak intensities. The flash fraction 16 (0.29 mg/mL DW) gave rise to the highest TPC, DPPH, and FRAP values, followed by the flash fractions ranked in the following order: 17 > 14 > 19 (Figure 2). This fact indicates that the most abundant antioxidant compounds in blackberry were of intermediate polarity. However, one fraction, i.e., fraction number 17 (0.54 mg/mL DW), with moderate antioxidant indices eluted in 80% methanol gradient, suggests the presence of low polarity antioxidants. An additional fraction, fraction 1, gave rise to a peak of low intensity, eluting in the high polar region of the flash chromatogram. However, these fractions exhibited a considerably lower TPC compared to the other fractions.

In addition to the satisfactory correlation among the flash chromatogram and the antioxidant indices of blackberries (Figure 2 and Figure 3), a strong positive correlation was also observed among the latter (Table 1). TPC and FRAP exhibited an *r* ≥ 0.9870, whereas a similar positive strong correlation was attained among TPC and DPPH values (*r* ≥ 0.9870). An F-C assay is used to estimate total phenolic content, but it shares a similar electron transfer reaction to that of DPPH^•^ and FRAP assays [16]. Thus, the regularly reported good correlations among these assays are not surprising. Our results corroborate with those of earlier studies demonstrating that FRAP [17] and DPPH^•^ [18] with TPC, but also DPPH^•^ and FRAP assays [17], are well correlated to each other.

### 2.2. Characterization of Phenolic Compounds by LC-Q-TOF-MS/MS and UPLC-TQD-MS/MS Analyses

To evaluate which phenolic compounds were responsible for the antioxidant activities observed, the selected flash fractions (13–22, 27, and 28) were subjected to LC-MS/MS analyses. Untargeted LC-Q-TOF-MS/MS analysis was employed as a profiling of phenolic compounds present in the crude blackberry extract. Subsequent targeted UPLC-TQD-MS/MS analysis was performed for confirmatory identification and quantification of the predominant phenolic compounds in these twelve flash fractions that possessed moderate to high antioxidant indices.

The predominant phenolic compounds detected in the LC-Q-TOF-MS/MS of blackberry crude extract are shown in Table 2, whereas the obtained LC-MS chromatogram is displayed in Appendix A with annotated peak numbers corresponding to the numbers listed in Table 2. The use of accurate LC-Q-TOF mass measurements (<5 ppm) led to the identification of thirty phenolic compounds, of which cinnamtannin A2 (ellagitannin) is being reported first time in blackberries. The identified phenolics can be classified into five distinctive classes, namely anthocyanins, proanthocyanidins, ellagitannins, flavonoids, and hydroxybenzoic acids. Fourteen phenolic compounds were identified with authenticated standards, while the additional phenolics were tentatively identified with the literature data on their MS/MS data (References column, Table 2).

The abundance of the identified compounds was confirmed using UPLC-TQD-MS/MS, as illustrated in the heat map of Figure 3. The compounds for which authentic standards were not commercially available were quantified as equivalents of other standards that shared common structural characteristics, as indicated in the footnotes of Figure 3. Although fraction 1 exhibited moderate antioxidant indices, the LC-Q-TOF analysis showed mainly the presence of organic acids and not phenolic compounds. Hence, this fraction was not assessed for its individual phenolic compounds. Cyanidin-3-*O*-glucoside constitutes the most abundant polyphenol present in blackberries as evident from other previous studies [2,3,4]. However, blackberries are also rich sources of ellagitannins [19], with two of them, namely lambertianin C and sanguiin-H6/lambertianin A, being present in two of the most antioxidant-potent fractions (fractions 16 and 17). As expected in most flash fractions that eluted close to each other, the same phenolic compounds were present in more than one fraction but at different concentrations. For instance, cyanidin-3-*O*-glucoside and cyanidin-3-*O*-rutinoside were present at quantifiable levels among all the twelve selected fractions, and in most of the fractions, they were present in substantially lower concentrations compared to the highly abundant fractions (fractions 16 and 17). As has been previously reported, this effect is ascribed to the limited resolving power of FLC separation [12]. Nevertheless, the separation of these compounds in certain fractions could enable their subsequent combination and enrichment to produce antioxidant-rich extracts for food and non-food applications.

**Table 2 molecules-28-01933-t002:** Characterization of phenolic compounds in lyophilized blackberry extract prior flash fractionation by LC-QTOF mass spectrometry.

Peak No.	Name	t_R_ (min.)	Molecular Formula	Observed [M-H]^−^ *m/z*	Calculated [M-H]^−^ *m/z*	Mass Error (ppm)	MS/MS Ions *m/z*	References
1	Gallic acid	0.93	C_7_H_6_O_5_	169.0143	169.0137	3.5	125.03; 111.03; 96.98; 155.03	[20,21] ^1^
2	Neochlorogenic acid	1.25	C_16_H_18_O_9_	353.0866	353.0873	−2.0	191.09; 155.05; 179.07	[22] ^1^
3	Catechin	2.21	C_15_H_14_O_6_	289.0723	289.0712	3.8	245.09; 203.08; 125.04	[22] ^1^
4	Protocatechuic acid	2.40	C_7_H_6_O_4_	153.0185	153.0188	−2.0	109.03; 91.02	[22] ^1^
5	Dihydromyricetin rhamnoside	2.48	C_21_H_22_O_12_	465.1015	465.1033	−3.9	125.03; 109.01; 241.06; 329.11	[22]
6	Cyanidin-3-*O*-glucoside	2.53	[C_21_H_21_O_11_]^+^	447.0929	447.0927	0.4	285.05; 269.06; 125.03	[20] ^1^
7	Cyanidin-3-*O*-rutinoside	2.66	[C_27_H_31_O_15_]^+^	593.1597	593.1506	−1.5	285.06; 284.05; 269.06; 287.07	[23,24] ^1^
8	Procyanidin trimer C type	2.85	C_45_H_38_O_18_	865.1987	865.1980	0.8	407.18; 289.14; 243.10	[25,26]
9	3-*p*-Coumaroylquinic acid	2.95	C_16_H_18_O_8_	337.0925	337.0923	0.6	173.05; 191.07	[27,28]
10	Procyanthocyanidin type	2.98	C_43_H_32_O_11_	723.1888	723.1866	3.0	289.15; 407.19; 525.26	[29]
11	Cyanidin-3-*O*-sophoroside	3.11	[C_27_H_31_O_16_]^+^	611.1586	611.1612	−1.2	285.06; 475.16; 241.07	[30]
12	Procyanidin B2	3.25	C_30_H_26_O_12_	577.1339	577.1346	−4.3	289.15; 407.18; 125.07; 408.19	[31,32] ^1^
13	Cinnamtannin A2	3.35	C_60_H_50_O_24_	1155.2719	1155.2771	0.6	577.3; 425.2; 407.2; 287.1; 451.2	[33] ^2^
14	3-Feruloylquinic acid	3.44	C_17_H_20_O_9_	367.1012	367.1029	−4.6	166.58; 193.02	[28,34]
15	Procyanidin trimer C type	3.75	C_45_H_38_O_18_	865.1976	865.1980	−0.5	287.13; 125.06; 407.18; 243.10	[25,26]
16	Epicatechin	4.31	C_15_H_14_O_6_	289.0726	289.0712	4.8	245.08; 203.06; 125.02; 179.04	[24,35] ^1^
17	Rutin	6.50	C_27_H_30_O_16_	609.1430	609.1456	−4.3	301.05; 271.04; 255.05; 151.02	[3] ^1^
18	Epigallocatechin	6.55	C_15_H_14_O_7_	305.0663	305.0661	0.7	219.11; 146.99; 225.92; 196.97	[36]
19	Lambertianin C isomer ^3^	6.87	[C_123_H_80_O_78_]^2-^	1401.1112	1401.1068	3.1	301.01; 633.09; 1250.64; 935.11	[37]
20	Sanguiin H-6/Lambertianin A	7.19	C_82_H_54_O_52_	1869.1470	1869.1503	−1.8	935.09; 1235.16; 1567.15; 633.08	[37]
21	Ellagic acid pentoside	8.09	C_19_H_14_O_12_	433.0403	433.0407	−0.9	300.98; 153.03; 78.98	[3]
22	Ellagic acid	8.14	C_14_H_6_O_8_	300.9998	300.9984	4.7	301.01; 117.05; 284.02; 229.03	[37] ^1^
23	Quercetin-3-*O*-galactoside	8.15	C_21_H_20_O_12_	463.0867	463.0877	−2.2	301.03; 271.02; 255.03	[3] ^1^
24	Quercetin-3-*O*-glucoside	8.46	C_21_H_20_O_12_	463.0866	463.0877	−2.4	301.06; 151.01; 271.05; 255.04	[3] ^1^
25	Quercetin-3-*O*-hexoside	8.68	C_21_H_20_O_12_	463.0865	463.0877	−2.0	301.05; 271.03; 151.02	[3]
26	Quercetin-3-pentoside	9.19	C_20_H_18_O_11_	433.0783	433.0771	2.8	301.06; 271.09; 154.04	[3]
27	Kaempferol-3-*O*-rutinoside	9.32	C_27_H_30_O_15_	593.1491	593.1506	−2.5	269.10; 285.09; 257.11; 225.11; 125.05	[3] ^1^
28	Quercetin	12.25	C_15_H_10_O_7_	301.0357	301.0348	3.0	178.05; 125.05; 137.05; 147.00	[37] ^1^

^1^ Confirmed with authentic standard; ^2^ the compound has not been previously reported in blackberries; ^3^ doubly charged ions [M-H]^2−^ dominant for this compound.

The mid-polar fraction 16, which exhibited the highest antioxidant indices (Figure 2) among all the different flash fractions, mainly contained (Figure 3) cyanidin-3-*O*-glucoside (31.79 μg/mL), followed by epicatechin (11.46 μg/mL), cyanidin-3-*O*-rutinoside (4.68 μg/mL), procyanidin B2 (1.04 μg/mL), and lambertianin C (0.59 μg/mL). Fraction 17 exhibited the second highest antioxidant potential and showed the presence of the same quantifiable compounds, except lambertianin C (MW = 2804 Da), at different concentrations. In particular, fraction 17 had higher cyanidin-3-*O*-glucoside (33.35 μg/mL) while the procyanidin B2 (0.63 μg/mL) had reduced by almost 40% compared to fraction 16. Instead of lambertianin C, fraction 17 had a lower molecular weight ellagitannin, namely sanguiin-H6/lambertianin A (MW = 1870 Da, 0.40 μg/mL). There were high contents of cyanidin-3-*O*-glucoside and cyanidin-3-*O*-rutinoside on either side (fractions 13–15 and 18–21) of flash fractions 16 and 17, and these two anthocyanins could be responsible for high antioxidant indices. Reyes-Carmona et al. [35] have also shown that there is a significant correlation among antioxidant activities, namely FRAP and oxygen radical absorbance capacity (ORAC), and the concentration of cyanidin-3-O-glucoside and cyanidin-3-*O*-rutinoside in blackberries. Similarly, Gong et al. [4], through boosted regression tree analysis, also showed that cyanidin-3-*O*-glucoside and chlorogenic acid were the main contributors in the antioxidant activity of free phenolics from different blackberry varieties. In the blackberry variety used in this study, chlorogenic acid was not one of the most abundant compounds. It has been suggested that ortho-diphenols exert higher antioxidant activity compared to simple phenols because of phenoxy-radical stabilization upon hydrogen bonding [37]. Hence, the ortho-dihydroxy moiety in the B ring of cyanidin [37,38], glucoside, and rutinoside could explain their high antioxidant potential of the produced fractions.

It was further evident from the obtained results (Figure 3) that the presence of ellagitannins, exclusively only in these two most potent fractions (16 and 17), together with catechins (procyanidin B2 and epicatechin), suggests their strong contributions in the antioxidant capacity of blackberries. Quercetin is also a flavonoid that contains an ortho 3′,4′-dihydroxy moiety in the B ring, as in cyanidins, but has shown lower scavenging activity compared to quercetin [39]. The higher potential of cyanidins compared to quercetin derivatives was also observed in the present study based on their contents in fractions 27 and 28. The findings suggest potential synergistic and antagonistic effects between the compounds and their concentrations that would affect the antioxidant indices [4]. Gangopadhyay et al. [12] have also suggested that, even though a number of phenolic compounds may be highly bioactive, their presence in substantially low concentrations in a sample may be the reason that they do not exert an evident antioxidative effect. Similarly, the predominant phenolics in a sample may exhibit low or no antioxidant activities. Therefore, it is important that both the concentration and the inherent bioactive properties of phenolics be taken into consideration for determining those with the highest potency in the examined samples. An additional factor that may affect the assessment of antioxidant potentials may be the presence of predominately non-phenolic compounds, which are reactive to the colorimetric and radical scavenging assays, such as the sugars that were mainly present in fraction 1 of this study [40]. As it has also been suggested in the past, both the number of hydroxyl (OH) groups and the presence of double bonds that are conjugated to OH and ketonic groups can highly affect the antioxidant capacity of certain compounds [41].

## 3. Materials and Methods

### 3.1. Samples and Reagents

Blackberries were purchased from a local store (SuperValu, Ireland). The berries originated from Morocco and belonged to the Elvira variety. After lyophilization of whole berries for 48 h on a freeze-dryer (Lyovapor™ L-300, Büchi, Flawil, Switzerland), the lyophilized blackberries were ground with a mixer, and the samples were further lyophilized for another 48 h to total dryness. The final lyophilized sample was pulverized into a fine powder of a particle size ≤400 μm using a mesh sieve, vacuum-packed, and stored at −20 °C and in the dark until further use.

F-C phenol reagent, gallic acid, sodium carbonate (Na_2_CO_3_), Trolox, formic acid (HCOOH), sodium chloride (NaCl), magnesium chloride hexahydrate (MgCl_2_•6H_2_O), and acetone (Ace) were purchased from Sigma-Aldrich (now Merck, Wicklow, Ireland). The standards of phenolic compounds were obtained from Extrasynthese (Extrasynthese Co., Genay Cedex, France). Acetonitrile (ACN) was acquired from Romil (Lennox Laboratory Supplies LTD, Dublin, Ireland). Milli-Q^®^ (18.2 mΩ) (Merck Millipore, Molsheim, France) water (H_2_O) was used throughout the experiments.

### 3.2. Extraction of Phenolic Compounds from Blackberries

The pulverized and lyophilized sample was extracted with 70% acetone at 60 °C for 60 min. The optimal conditions were based on an in-house solid–liquid extraction method for acquiring phenolic-rich blackberry extracts from our previous blackberry project (*Cardio Rubus)* as well as the findings by Boeing et al. (2014), where the 70% acetone was the most efficient solvent in extracting blackberry polyphenols [42]. The sample extraction was performed at a 1:10 ratio (30 g lyophilized blackberry in 300 mL of solvent) as determined by the study of Boeing et al. [42] with a shaker (MAXQ 8000, Thermo Scientific) set at 100 rpm. Upon extraction, the sample was vacuum filtered, and the remaining pellet was washed from the container with 100 mL of 70% acetone and re-filtered. Subsequently, the filtrate was placed in a round-bottomed flask for drying in a rotary evaporator (Büchi, Flawil, Switzerland) with a water bath set at 40 °C and reducing pressure until the entire removal of acetone was achieved. Finally, after collection of the H_2_O fraction, the remaining extract in the flask was washed with Milli-Q H_2_O, and after merging with the initial H_2_O fraction, it was stored at −80 °C until further lyophilization (Lyovapor™ L-300, Büchi, Flawil, Switzerland).

### 3.3. Flash Chromatography

The FLC fractionation was performed with an in-house method. An amount of 0.5 g of lyophilized blackberry extract was added to 4.5 g of C18 silica sorbent powder and mixed to a slurry after addition of methanol (MeOH) to achieve homogenous distribution of the extract in the sorbent. The mixture of blackberry extract with the C18 sorbent was left to dry at ambient temperature, and it was subsequently added and tightly packed into a loading cartridge for FLC fractionation (IntelliFlash, Modell 310, Varian, CA, USA). The flash column used was an Agilent, C18, AX1409-1, SF10 (column size of 10 g), and a binary solvent system was employed comprising H_2_O and MeOH. A stepwise gradient from 0 to 100% B (0% B for 5 min, 10% B from 5–10 min, 20% B from 10–15, 30% B from 15–20, 80% Β from 20–25, and 100% B from 25–32) and a flow rate of 10 mL/min were set to separate the phenolic compounds of the extract. The fractions were monitored at four different wavelengths, namely 280 nm, 320 nm, 360 nm, and 399 nm (Figure 1), and fractions were collected every minute. The schematic flow chart illustrating the previously described extraction method and FLC fractionation, as well as the analysis of the phenolic-rich blackberry extracts, is illustrated in Figure 4.

### 3.4. Determination of TPC with the F-C Assay

TPC was estimated using the F-C phenol method, as described by the modified method of Singleton et al. [43]. Aliquots of 100 μL of the appropriately diluted extract/standard/blank, 100 μL of H_2_O, 100 μL of F-C reagent, and 700 μL of 20% Na_2_CO_3_ were added together and vortexed. Soon after vortexing, the reaction mixture was placed for 20 min in the dark and subsequently centrifuged at 13,000 rpm for 3 min. The absorbance of 200 μL of the supernatant was measured at 735 nm using an EPOCH2 Plate Reader (BioTek, Winooski, VT, USA) on a polystyrene microplate. All measurements were performed in triplicate for each sample and standard solution, and the samples were corrected after subtraction of the reagent blank. Gallic acid solutions were used as standards and the TPC was expressed as mg gallic acid equivalents (GAE)/100 g freeze-dried extract.

### 3.5. Ferric Reducing Antioxidant Power (FRAP) Assay

The FRAP assay was conducted based on the method described by Stratil et al. [44] with slight modifications. The FRAP reagent was freshly prepared before each experiment and contained 38 mM of CH_3_COONa in Milli-Q^®^ H_2_O (pH 3.6), 10 mM of TPTZ in 40 mM HCl, and 20 mM of FeCl_3_•6H_2_O in Milli-Q^®^ H_2_O in a ratio equal to 10:1:1. The reagent was incubated for 5 min prior to analysis at 37 °C. In a polystyrene microplate, 20 μL of appropriately diluted sample/standard/blank and 180 μL of FRAP reagent were added, mixed, and incubated for 40 min at 37 °C. At the end of the incubation period, the absorbance was determined immediately by an EPOCH2 Plate Reader (BioTek, Winooski, VT, USA) at 593 nm in a polystyrene plate. Trolox solutions were used as standards and FRAP values were expressed as mg trolox equivalents (TE)/100 g freeze-dried extract.

### 3.6. 2,2′-Diphenyl-1-picrylhydrazyl Radical (DPPH^•^) Assay

DPPH^•^-scavenging capacity was determined as described by Goupy et al. [45] with slight modifications. DPPH^•^ solution was freshly prepared before each experiment after the dissolution of DPPH^•^ in MeOH (0.238 mg/mL). Following US for 30 min in the dark, the stock solution was stored for 2 hours in the dark and under refrigeration to ensure that all DPPH^•^ had been dissolved and stabilized prior to analysis. Subsequently, 100 μL of the working solution (1:5 diluted stock solution with MeOH) was mixed with 100 μL of properly diluted sample/standard in a polystyrene microplate. Trolox solutions were used as standards; for the control sample 100 μL of 80% MeOH was added in place of the standard extract. The mixture was kept at room temperature and in the dark for 30 min, and then the absorbance was measured at 515 nm on EPOCH2 Plate Reader (BioTek, Winooski, VT, USA) using the extraction solvent as blank. The radical scavenging capacity was expressed as mg TE/100 g freeze-dried extract.

### 3.7. Characterization of Phenolic Compounds by LC-QTOF-MS/MS Analysis

Extensive characterization of the phenolic compounds present in the lyophilized crude extract of blackberry extract prior to FLC was carried out using the LC-gradient method of Xu et al. [46] with modifications. An Alliance 2695 HPLC system (Waters Corporation, Milford, MA, USA) coupled with a Q-TOF Premier mass spectrometer were employed and the chromatographic separation of the analytes was carried out at 40 °C using an Atlantis T3 C18 column (100 × 2.1 mm, 3.0 μm particle size). A binary mobile phase comprising 0.1% HCOOH in H_2_O (Solvent A) and 0.1% HCOOH in ACN (solvent B) was used, with a stepwise gradient from 10 to 90% solvent B (0–1 min, 10–10% B; 1–4 min, 10–20% B; 4–10 min, 2–30% B; 10–15 min, 30–40% B; 15–18 min, 40–50% B; 18–22 min, 50–75% B; 22–23 min, 75–85% B; 23–24 min, 85–100% B; and 25–25min, 100–10%) for a total run time of 25 min and at a flow rate of 0.3 mL/min [45]. ESI mass spectra were recorded in negative ion mode [M-H^-^] for a mass range (*m*/*z*) 100–1500. Argon was used as collision gas (12–20 eV) to obtain collision-induced dissociation (CID) mass spectra. The MS data were centroided and corrected throughout acquisition using an external reference of leucine enkephalin [M-H^−^] (*m*/*z* 554.2615) at a flow rate of 12 mL/min.

### 3.8. Quantification of Phenolic Compounds through UPLC-TQD-MS/MS Analysis

UPLC-TQD-MS/MS for the quantification of the most abundant phenolic compounds in all the phenolic-rich fractions of the blackberry extract was performed with a Waters Acquity UPLC (Waters Corporation, Milford, MA, USA) coupled with a tandem quadrupole detector (TQD) after adapting the previous method of Gangopadhyay et al. [12]. An Acquity UPLC HSS T3 (100 × 2.1 mm; 1.8 μm particle size) was employed, utilizing a binary mobile phase of 0.1% aqueous formic acid (solvent A) and 0.1% formic acid in ACN (solvent B) and at a flow rate of 0.5 mL/min for 10 min. An MRM method was used to acquire quantitative data. The MRM transitions for each compound were obtained through the Waters Intellistart^TM^ software (Waters Corporation, Milford, MA, USA) as illustrated in Appendix A. UPLC-TQD-MS/MS data were obtained in negative ESI mode for all the phenolic compounds except for cyanidin-3-*O*-glucoside and cyanidin-3-*O*-rutinoside, for which positive ESI was employed. The quantification of phenolic compounds in the sample and standards was carried out using the TargetLynx^TM^ software (Waters Corporation, Milford, MA, USA).

### 3.9. Statistical Analysis

FLC fractionation was performed on only one extract, and each phenolic-rich fraction was analyzed three times for its TPC, DPPH, and FRAP indices. However, a coefficient of variation less than 15% was deemed permissible for the technical replicates of spectrophotometric measurements. The heat map for the quantified phenolic compounds from the FLC fractions was generated through the conditional formatting of Excel 16.31 (Microsoft Corp., Redmond, WA, USA). The same software was also used to estimate the Pearson correlation coefficients ® among the different antioxidant indices employed.

## 4. Conclusions

Flash chromatography is a robust and rapid separation method to identify antioxidant polyphenols and aid immensely in the bioactivity-guided fractionation studies. In blackberries, the flash fractions exhibited a strong positive correlation (*r* ≥ 0.986) with their phenolic contents and antioxidant indices. The role of anthocyanins, cyanidin-3-*O*-glucoside, and cyanidin-3-*O*-rutinoside toward antioxidant activity is further confirmed. More importantly, this study showed that other polyphenols, ellagitannins (lambertianin C isomer and sanguiin H-6/lambertianin A), and catechins (procyanidin B2 and epicatechin) contribute synergistically to the antioxidant capacity of blackberries, as they were found higher or only in the most potent flash fractions of blackberry extract. Flash chromatography fractionation of blackberry extracts or similar matrices using food-compatible solvents such as water and ethanol or acetone can be a viable way in the production of phenolic-rich extracts for food and non-food applications.

## Figures and Tables

**Figure 1 molecules-28-01933-f001:**
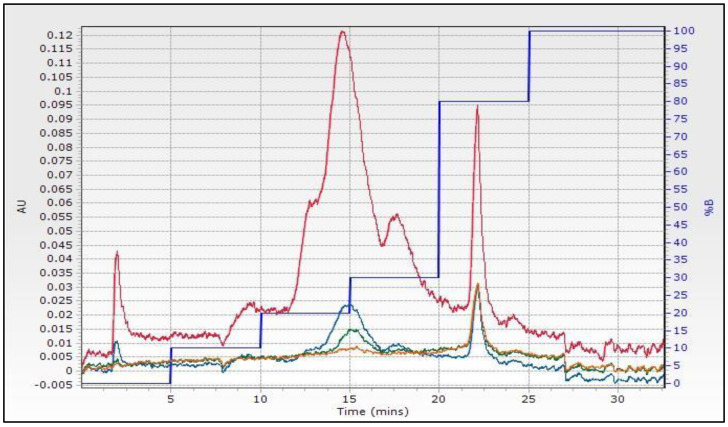
Flash chromatograms obtained following fractionation of the phenolic extract of lyophilized blackberries. Different colors of chromatographic peaks represent different wavelengths, namely 280 nm (Red), 320 nm (Blue), 360 nm (Green), and 399 nm (Yellow). Dark blue line shows the % methanol (%B) gradient.

**Figure 2 molecules-28-01933-f002:**
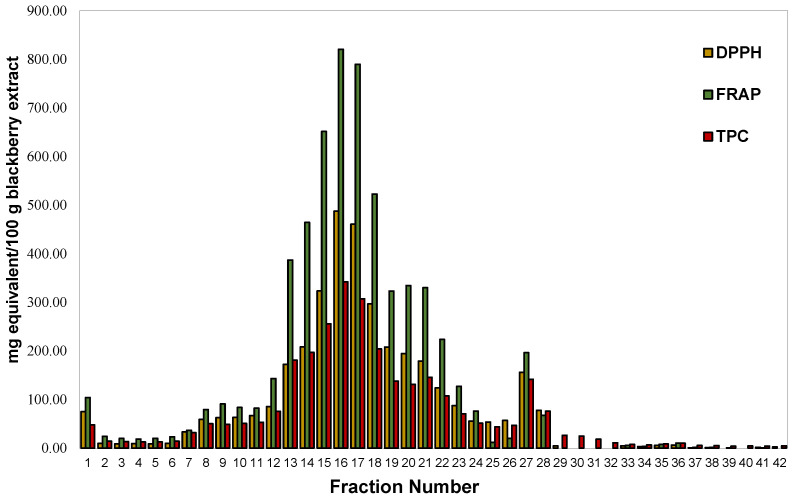
TPC (in gallic acid equivalent), DPPH, and FRAP both in Trolox equivalent of the FLC fractions of blackberry extract.

**Figure 3 molecules-28-01933-f003:**
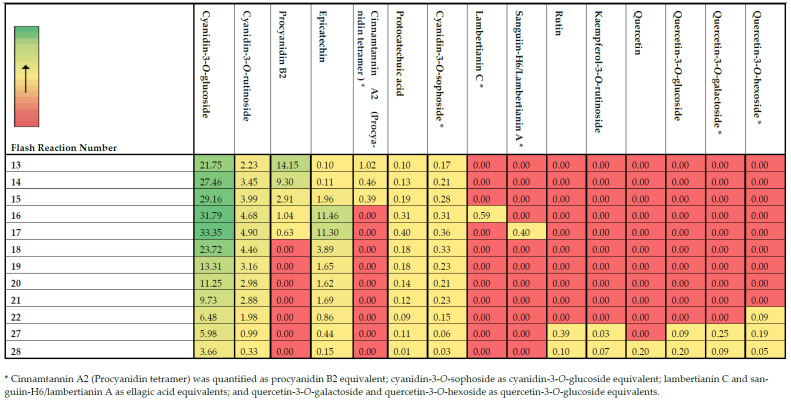
Heat map of the quantified major phenolic compounds in the most antioxidant-potent fractions through UPLC-TQD-MS/MS. Data are expressed as μg/mL.

**Figure 4 molecules-28-01933-f004:**
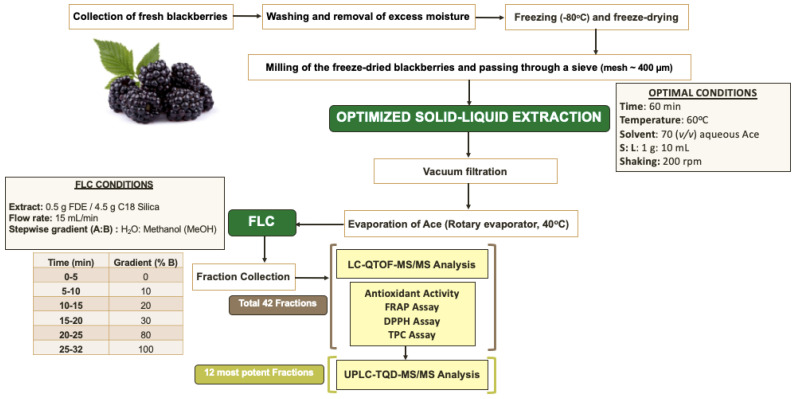
Schematic flow chart illustrating the optimized extraction process, flash fractionation, and analysis of the blackberry phenolic-rich fractions. Ace = acetone, S = solid, L = liquid, FDE = freeze dried extract.

**Table 1 molecules-28-01933-t001:** Correlation (*r*) between total polyphenols (TPC) and antioxidant activity (DPPH and FRAP assays) for the blackberry extract fractions.

	*DPPH*	*FRAP*	*TPC*
**DPPH**	1.000		
**FRAP**	0.986	1.000	
**TPC**	0.987	0.987	1.000

## Data Availability

Data is contained within the article or Appendix A.

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
