# Peer review of "Antioxidant Guided Fractionation of Blackberry Polyphenols Show Synergistic Role of Catechins and Ellagitannins"

_molecules, 2023, doi:10.3390/molecules28041933_

Round 1
Reviewer 1 Report
The recovery of two potent fractions (16 and 17) in the acetone extract of blackberry should be provided.
Author Response
Reviewer 1
The recovery of two potent fractions (16 and 17) in the acetone extract of blackberry should be provided.
Response: The recovery of the two potent flash fractions has been provided (Lines 96 and 99).
Reviewer 2 Report
The work is relevant, however, it lacks novelty
Introduction: I believe that this work would benefit with an introduction more complete, with a complete explanation and with several published works and translating the importance of the synergy of several antioxidants in foods.
Material and methods: using good documentary resources, although a bit confused in the way experimental data were revealed
Conclusions: based on the data and references used
Author Response
Reviewer 2
We are extremely grateful to the reviewer for the constructive criticism of our manuscript,. We have addressed those in the revised version.
The work is relevant, however, it lacks novelty
Response: We have applied bioactivity guided flash fractionation of antioxidant polyphenols. As far as we are aware that this is the first flash chromatography fractionation study of antioxidant polyphenols in blackberries. Past literature on the fractionation and enrichment of blackberry polyphenols have almost always associated that the most abundant blackberry polyphenols (i.e. anthcyanins) to their antioxidant capacity. Our flash fractionation study shows that the catechins and ellagitannins also contribute towards the antioxidant activity.
Introduction: I believe that this work would benefit with an introduction more complete, with a complete explanation and with several published works and translating the importance of the synergy of several antioxidants in foods.
Response: We have added the following text in the introduction from the previous antioxidant studies on blackberry polyphenols:
“Various separation methods to enrich blackberry polyphenols and link to their antioxi-dant activity have been reported. Acosta et al. [6] separated blackberry polyphenols using membrane filtration of different molecular weight cut off (MWCO) sizes (1–150 kDa), where the authors found membrane with MWCO of 2 kDa as the best for fractionation of anthocyanins. Ghosh et al. [7] also applied membrane filtration sequentially (50 kDa fol-lowed by 300 Da nanofiltration) to successfully concentrate antioxidant anthocyanins from Indian blackberry (Syzygium cumini) juice. Elisia et al. [8] on the other hand applied gel filtration to obtain an anthocyanin-enriched extract of blackberry (Rubus fruticosus sp). The total anthocyanin content following gel filtration was 20-fold higher than the original crude extract, while the antioxidant activity was enhanced by 7-fold for the anthocya-nin-enriched fraction. Sánchez‑Velázquez et al. [9] used column chromatography em-ploying different solvents, and adsorbent and ion-exchange resins (i.e. Amberlite XAD-7 resin followed by Sephadex LH-20 resin) to concentrate ellagitannins from Mexican wild blackberries. Cho et al. [10] applied semi-preparative HPLC to fractionate blackberry fla-vonols and interestingly showed that the flavonols had minor (<5%) contribution towards the total antioxidant capacity in blackberries. In the present study, flash chromatography is employed for the first time to fractionate blackberry polyphenols and associate them to their antioxidant activity.”
Material and methods: using good documentary resources, although a bit confused in the way experimental data were revealed
Response: We have revised the material and methods, especially the sample preparation section.
Conclusions: based on the data and references used
Response: Conclusions were made based on our experimental data.
Reviewer 3 Report
This study was focused on the assessment of a systematic fractionation approach employing flash chromatography to determine the most potent fractions of blackberry phenolic extract. The research is interesting and could help to efficiently make use of blackberry phenolics for natural antioxidants. However, revisions are still needed, questions and suggestions are as following.
1. Generally, acidified ethanol or methanol solution was applied for extraction phenoilcs from fruits containing abundant anthocyanins. 70% aqueous acetone was used in this stuy, why?
2. LC-MS/MS, including LC-Q-TOF-MS/MS and UPLC-TQD-MS/MS, was suggested to be defined for the first time.
3. Introduction section is not enough for full y introducing search backgrounds.
4. the identifications of thirty phenolic compounds in the extracts was not clear. please refer to some newly published paper related to berry phenolics, DOI: 10.1016/j.lwt.2022.114308. The identification process, especially the firstly identified found compounds including cinnamtannin A2 (ellagitannin) and procyanidin tetramer B type, should be provided. And other tentatively identified compounds could provide the references.
5. In table2, apigenin was listed. Was this flavonoid compound identified by its standard? Apigenin is a weak polar phenolics, generally eluted in the back end of chromatograms, please confirm.
6. In section 3.7, for characterization of phenolic compounds, the same LC-QTOF-MS was applied in the similar research, and the reference DOI: 10.1016/j.foodchem.2019.05.058 is more suitable.
7. In section 3.1, for whole berries containing high sugar, it’s difficulty to dry, and more difficulty to be pulverized before fully dried. How to deal with your samples by your description of “After lyophilisation as whole berries for 48 h (Lyovapor™ L-300, Büchi, Flawil, Switzerland), blackberries were pulverized and lyophilized for another 48 h to total dryness.” How do you know the particle size ≤ 400 μm?
Author Response
Reviewer 3
Response: We are extremely grateful to the reviewer for the constructive criticism of our manuscript. The in depth knowledge of the reviewer has helped to strengthen our manuscript greatly.
This study was focused on the assessment of a systematic fractionation approach employing flash chromatography to determine the most potent fractions of blackberry phenolic extract. The research is interesting and could help to efficiently make use of blackberry phenolics for natural antioxidants. However, revisions are still needed, questions and suggestions are as following.
- Generally, acidified ethanol or methanolsolution was applied for extraction phenoilcs from fruits containing abundant anthocyanins. 70% aqueous acetone was used in this study, why?
Response: We agree with the reviewer on general used of acidifiedalcohols. However, in our own previous work on blackberries (CardioRubus project) and the observation made by Boeing et al. 2014 (reference # 41) showed that 70% acetone was the most efficient in extracting blackberry polyphenols. We have amended the sentence below to support the solvent choice.
“The optimal conditions were based on an in-house solid-liquid extraction method for ac-quiring phenolic-rich blackberry extracts from our previous blackberry project (Cardio-Rubus) as well as the findings by Boeing et al. (2014), where the 70% acetone was the most efficient solvent in extracting blackberry polyphenols [41].” (Line 211-213).
- LC-MS/MS, including LC-Q-TOF-MS/MS and UPLC-TQD-MS/MS, was suggested to be defined for the first time.
Response: As for the qualitative analysis for polyphenols using LC-Q-TOF-MS/MS, we have our own in-house method first developed in 2010 (Hossain et al/ 2010). However, when we applied the gradient used by Xu et. al. 2020 (reference # 47), the chromatographic peak separations were much more resolved than our in-house method. Therefore, we adapted the gradient described by Xu et. al 2020 for the LC-QTOF-MS/MS in this study. For the UPLC-TQD-MS/MS analysis, we used the same LC-gradient as described by Gangopadhyay (reference #12), where the MRM transitions for additional phytochemicals were applied based on the Waters Intellistart® software for the commercially available standards or the MS/MS data from QTOF. Both references are cited in this manuscript.
- Introduction section is not enough for fully introducing search backgrounds.
Response: Introduction is revised and following sentences have been added to set the backgrounds:
“On the other hand. Sariburun et al. [5] also found that the antioxidant activities of blackberries were highly correlated (0.93 ≤ r ≤ 0.99) with their anthocyanin contents, while the total flavonoid content was relatively less correlated with antioxidant activity (0.85 ≤ r ≤ 0.89).
Various separation methods to enrich blackberry polyphenols and link to their antioxidant activity have been reported. Acosta et al. [6] separated blackberry polyphenols using membrane filtration of different molecular weight cut off (MWCO) sizes (1–150 kDa), where the authors found membrane with MWCO of 2 kDa as the best for fractionation of anthocyanins. Gosh et al. [7] also applied membrane filtration sequentially (50 kDa fol-lowed by 300 Da nanofiltration) to successfully concentrate antioxidant anthocyanins from Indian blackberry (Syzygium cumini) juice. Elisia et al. [ 8] on the other hand applied gel filtration to obtain an anthocyanin-enriched extract of blackberry (Rubus fruticosus sp). The total anthocyanin content following gel filtration was 20-fold higher than the original crude extract, while the antioxidant activity was enhanced by 7-fold for the anthocyanin-enriched fraction. Sánchez‑Velázquez et al. [9] used column chromatography em-ploying different solvents, and adsorbent and ion-exchange resins (i.e. Amberlite XAD-7 resin followed by Sephadex LH-20 resin) to concentrate ellagitannins from Mexican wild blackberries. Cho et al. 2005 applied semi-preparative HPLC to fractionate blackberry flavonols and interestingly showed that the flavonols had minor (<5%) contribution to-wards the total antioxidant capacity in blackberries [10]. In the present study, flash chromatography is employed for the first time to fractionate blackberry polyphenols and associate them to their antioxidant activity.”
- the identifications of thirty phenolic compounds in the extracts was not clear. please refer to some newly published paper related to berry phenolics, DOI: 10.1016/j.lwt.2022.114308. The identification process, especially the firstly identified found compounds including cinnamtannin A2 (ellagitannin) and procyanidin tetramer B type, should be provided. And other tentatively identified compounds could provide the references.
Response: We have added the following text to clarify the process of identification. We thank the reviewer for providing a reference that helped us to expand our identification process for the two ellagitannins firstly identified in blackberries.
The compound eluting at 3.30 min. (peak 14, Table 2) showed m/z 1153.2599 [M-H]-, and the elemental composition analysis predicted the chemical formula as C60H50O24. corresponded to
- In table2, apigenin was listed. Was this flavonoid compound identified by its standard? Apigenin is a weak polar phenolics, generally eluted in the back end of chromatograms, please confirm.
Response: Thank you for the thorough scrutiny of the LC-QTOF data. There was no apigenin in this blackberry extract, and we have removed it from the Table 2.
We fully agree with the reviewer that apigenin is a weak polar flavonoids and should not be eluting at the polar end of the reversed LC chromatogram.
- In section 3.7, for characterization of phenolic compounds, the same LC-QTOF-MS was applied in the similar research, and the reference DOI: 10.1016/j.foodchem.2019.05.058 is more suitable.
Response: We thank authors for the suggestion of suitable LC-QTOF method. In fact, for almost all untargeted analysis for polyphenols using LC-Q-TOF-MS/MS, we have our own in-house method first developed in 2010 (Hossain et al. 2010), and we adapted this in-house method for various matrices. However, when we applied the gradient used by Xu et. al. 2020(reference # 47), the chromatographic peak separations were much more resolved for blackberry extract than our in-house method. Therefore, we adapted the gradient described by Xu et. al. 2020 for the LC-QTOF-MS/MS in this study.
- In section 3.1, for whole berries containing high sugar, it’s difficulty to dry, and more difficulty to be pulverized before fully dried. How to deal with your samples by your description of “After lyophilisation as whole berries for 48 h (Lyovapor™ L-300, Büchi, Flawil, Switzerland), blackberries were pulverized and lyophilized for another 48 h to total dryness.” How do you know the particle size ≤400 μm?
Response: We have clarified the sample preparation in section 3.1 with following text:
“After lyophilisation of whole berries for 48 h on a freeze-dryer (Lyovapor™ L-300, Büchi, Flawil, Switzerland), the lyophilized blackberries were pulverized grounded with a mixer, and the samples was further lyophilized for another 48 h to total dryness. The final lyophilized sample was groundedpulverized into a fine powder of a particle size ≤ 400 μm using a mesh sieve, vacuum-packed and stored at -20 oC and in the dark until further use.”
Reviewer 4 Report
I reviewed the article entitled " Antioxidant guided fractionation of blackberry polyphenols show synergistic role of catechins and ellagitannins", written by Katerina Tzima, Gontorn Putsakum and Dilip K. Rai.
The purpose of this manuscript is to characterize the most powerful phenolic fractions of blackberry extract using a systematic fractionation approach with flash chromatography. In addition, the authors propose a bioactive purification method that could make them useful in food applications or in the phyto-pharmaceutical industry.
The scientific collect is very interesting, and each paragraph is described exhaustively. The results are clear and support the hypothesis of the work. The methods are well described. The conclusions contain the key message of the work.
Author Response
Reviewer 4
I reviewed the article entitled " Antioxidant guided fractionation of blackberry polyphenols show synergistic role of catechins and ellagitannins", written by Katerina Tzima, Gontorn Putsakum and Dilip K. Rai.
The purpose of this manuscript is to characterize the most powerful phenolic fractions of blackberry extract using a systematic fractionation approach with flash chromatography. In addition, the authors propose a bioactive purification method that could make them useful in food applications or in the phyto-pharmaceutical industry.
The scientific collect is very interesting, and each paragraph is described exhaustively. The results are clear and support the hypothesis of the work. The methods are well described. The conclusions contain the key message of the work.
Response: We are grateful to the reviewer for the positive kind words, which shall encourage our research team to bring more exciting research findings.